# Biogenic Ferrihydrite Nanoparticles: Synthesis, Properties In Vitro and In Vivo Testing and the Concentration Effect

**DOI:** 10.3390/biomedicines9030323

**Published:** 2021-03-22

**Authors:** Sergey V. Stolyar, Oksana A. Kolenchukova, Anna V. Boldyreva, Nadezda S. Kudryasheva, Yulia V. Gerasimova, Alexandr A. Krasikov, Roman N. Yaroslavtsev, Oleg A. Bayukov, Valentina P. Ladygina, Elena A. Birukova

**Affiliations:** 1Federal Research Center KSC SB RAS, Kirensky Institute of Physics, 660036 Krasnoyarsk, Russia; stol@iph.krasn.ru (S.V.S.); jul@iph.krasn.ru (Y.V.G.); kaa3000@yandex.ru (A.A.K.); yar-man@bk.ru (R.N.Y.); helg@iph.krasn.ru (O.A.B.); 2Krasnoyarsk Scientific Center, Federal Research Center KSC SB RAS, 660036 Krasnoyarsk, Russia; lampa15@bk.ru (A.V.B.); ladvp@mail.ru (V.P.L.); 3Biophysics Department, Siberian Federal University, 660041 Krasnoyarsk, Russia; n-qdr@yandex.ru; 4Federal Research Center KSC SB RAS, Scientific Research Institute of Medical Problems of the North, 660022 Krasnoyarsk, Russia; helena.biryukova.1996@gmail.com; 5Federal Research Center KSC SB RAS, Institute of Biophysics, 660036 Krasnoyarsk, Russia

**Keywords:** ferrihydrite nanoparticles, concentration effect, microorganisms *Klebsiella oxytoca*, neutrophilic granulocytes, chemiluminescence, toxicology

## Abstract

Biogenic ferrihydrite nanoparticles were synthesized as a result of the cultivation of *Klebsiella oxytoca* microorganisms. The distribution of nanoparticles in the body of laboratory animals and the physical properties of the nanoparticles were studied. The synthesized ferrihydrite nanoparticles are superparamagnetic at room temperature, and the characteristic blocking temperature is 23–25 K. The uncompensated moment of ferrihydrite particles was determined to be approximately 200 Bohr magnetons. In vitro testing of different concentrations of ferrihydrite nanoparticles for the functional activity of neutrophilic granulocytes by the chemiluminescence method showed an increase in the release of primary oxygen radicals by blood phagocytes when exposed to a minimum concentration and a decrease in secondary radicals when exposed to a maximum concentration. In vivo testing of ferrihydrite nanoparticles on Wister rats showed that a suspension of ferrihydrite nanoparticles has chronic toxicity, since it causes morphological changes in organs, mainly in the spleen, which are characterized by the accumulation of hemosiderin nanoparticles (stained blue according to Perls). Ferrihydrite can also directly or indirectly stimulate the proliferation and intracellular regeneration of hepatocytes. The partial detection of Perls-positive cells in the liver and kidneys can be explained by the rapid elimination from organs and the high dispersion of the nanomaterial. Thus, it is necessary to carry out studies of these processes at the systemic level, since the introduction of nanoparticles into the body is characterized by adaptive-proliferative processes, accompanied by the development of cell dystrophy and tension of the phagocytic system.

## 1. Introduction

Ferrihydrite Fe_2_O_3_∙nH_2_O or oxyhydroxide Fe^3+^, in comparison with Fe^3+^ hydroxides and oxides, is the compound with the highest metastability. Due to this, ferrihydrite plays a huge role in the metabolism of living organisms. It is formed in the core of the protein complex named ferritin, which is a capsule of the protein apoferritin. The size of the ferrihydrite nanoparticles is typically in a narrow range from 2 to 8 nm. With increasing particle size, the Fe_2_O_3_∙nH_2_O → hematite transformation occurs [1]. The ability of enterobacteria *Klebsiella oxytoca*, isolated from pyrite deposits, to ferment iron citrate under anaerobic conditions and to form a trivalent metal hydrogel was first reported in [2]. The hydrogel is a secretory exopolysaccharide that contains galactose, glucuronic acid, and rhamnose, and is associated with nanoparticles of ferrihydrite [3,4,5]. Ferrihydrite nanoparticles synthesized by *Klebsiella oxytoca* can be a very promising drug for various applications and, therefore, they require comprehensive study. Of particular interest are the magnetic properties of nanoparticles of ferrihydrite. Although ferrihydrite is characterized by an antiferromagnetic ordering with a Néel temperature of ~350 K, the presence of defects caused by the nanoscale state leads to the appearance of the uncompensated magnetic moment in an antiferromagnetic particle of a magnitude reaching hundreds of Bohr magnetons [6,7,8,9]. Another important property of ferrihydrite nanoparticles synthesized by *Klebsiella oxytoca* is the presence of a polysaccharide capsule [4,10,11,12,13]. Pharmaceutical iron–polysaccharide complexes for the treatment of iron deficiency anemia consist of iron hydroxide nanoparticles (usually in the form of β-FeOOH) of a few nanometers in size. As polysaccharides, polymaltose, dextrans, etc. are used. All these properties open up prospects for the use of ferrihydrite nanoparticles in biomedical research [10,14], in which iron-containing complexes coated with polysaccharides are most often and widely used.

However, when studying the effect of ferrihydrite nanoparticles on living organisms, it should be borne in mind that nanoparticles have the ability to adsorb proteins from biological fluids and form a protein layer called the protein corona [15,16,17,18]. The ability of a particle to adsorb proteins and the composition of the protein corona depends on the physical properties of the nanoparticles, size, shape, and surface chemistry. The presence of a protein corona can significantly affect the properties of nanoparticles (hydrodynamic diameter, aggregation, solubility), the body’s immune response, the structure of adsorbed proteins, and the biological functions of attached molecules [15,16,17,18,19,20,21]. Surface biotransformation of nanoparticles unpredictably changes their overall pharmacological and toxicological profile, as well as their potential therapeutic or diagnostic functionality [15,16,22,23,24,25]. The protein corona can “screen” molecules on the surface of nanoparticles and cause a loss of specificity during targeting [22,25]; enzymes bound to the surface of nanoparticles can change their activity [23,24]. The process of formation of a protein crown on biogenic ferrihydrite nanoparticles has not yet been studied, although there are studies of the interaction of synthetic ferrihydrite with plasma proteins [26] or other magnetic nanoparticles coated with polysaccharides [27].

Thus, the prospect of the practical use of nanoparticles of ferrihydrite requires the study of the mechanisms of interaction of nanoparticles in different concentrations with cells of the body, the ways of their transformation and excretion, as well as possible toxic effects depending on the concentration of nanoparticles.

This work aims to study the physical (IR spectroscopy, Mössbauer spectroscopy, magnetometry) and biological properties in vitro and in vivo of biogenic ferrihydrite nanoparticles prepared as a result of the cultivation of microorganisms *Klebsiella oxytoca*.

## 2. Materials and Methods

### 2.1. Production Technology

The microorganisms *Klebsiella oxytoca* used were isolated from the sapropel of Lake Borovoye (Krasnoyarsk Territory). The sapropel selected in the lake was passed through a magnetic separator. Microorganisms grew on nutrient media containing various forms of iron (Fe^2+^ oxalate and Fe^3+^ citrate). Microorganisms of the *Klebsiella oxytoca* culture were plated on Lovley medium [28] on Fe^2+^ oxalate, which was used as a chelator, with the following composition (g/l): NaHCO_3_—2.5; CaCl_2_∙H2O—0.1; KCl—0.1; NH_4_Cl—1.5; NaH_2_PO_4_∙H_2_O—0.6; Mohr’s salt (FeSO_4_∙(NH_4_)_2_SO_4_∙6H_2_O)—0.4; oxalic acid (H_2_C_2_O_4_)—0.22; yeast extract 0.05 g/L. Microorganisms of the *Klebsiella oxytoca* culture were also grown in Lovley medium on iron citrate with a concentration of 0.5 g/l with the following composition (g/L): NaHCO_3_—2.5; CaCl_2_∙H20—0.1; KCl—0.1; NH_4_Cl—1.5; NaH_2_PO_4_∙H_2_O—0.6; yeast extract 0.05 g/L. Sampling was carried out 3–28 days after the inoculation of microorganisms in a nutrient medium. The time dependences of the change in the dry weight of bacteria grown on Fe^2+^ oxalate and Fe^3+^ citrate were obtained (Figure 1). Thus, the dynamic of bacterial sedimentation was observed over time and the influence of the cultivation regime on the growth of the bacterial culture was monitored.

To isolate ferrihydrite from the sediment and obtain a sol, the bacterial biomass was separated from the supernatant by centrifugation (10 min, 10,000 rpm), then the bacteria cells were destroyed by ultrasound (ultrasonic disintegrator UZDN (1 min, 44 kHz, 20 W)) 3 times for 3 min in distilled water at intervals of 10 min. Next, the resulting precipitate to remove fatty acids was sonicated and incubated in acetone for 30 min and washed with distilled water (sonication, centrifugation). Then, the precipitate was placed in a 2% NaOH solution, the incubation time was 1 h. The resulting precipitate was washed with distilled water with the addition of NaCl (sonication, centrifugation) until a pH of 8 supernatant was reached. Thus, a sol was obtained, which was subsequently dried at room temperature.

### 2.2. FTIR Spectroscopy

A qualitative analysis of the infrared spectra of ferrihydrite nanoparticles was carried out to determine the functional groups. The infrared spectra of ferrihydrite nanoparticles were measured on a VERTEX 80V (BRUKER, Germany, Karlsruhe) vacuum Fourier spectrometer in the mid-IR region of 400–4000 cm^−1^.

### 2.3. Mossbauer Spectroscopy

Mössbauer measurements were carried out on an MS-1104Em spectrometer (Kordon, Russia, Rostov-on-Don) with a Co^57^(Cr) source with a line width at half maximum of 0.24 mm/s on an absorber made of sodium nitroprusside powder. The thickness of the samples was 5–10 mg/cm^2^ based on the natural iron content, at which the intensities of the spectrum lines are linearly dependent on the iron content in the phase. Isomer chemical shifts are indicated relative to α-Fe.

### 2.4. Magnetometric Measurements

Magnetic measurements were carried out on a vibration magnetometer [29]. The test powder was fixed in a measuring capsule in paraffin. The data were corrected by subtracting the diamagnetic signal from the paraffin capsule. The temperature dependences of the magnetization were measured in cooling modes without an external magnetic field (ZFC—zero-field cooling) and in the field (FC—field cooling). Magnetization hysteresis loops M(H) were measured under ZFC conditions. For comparison, the paper presents the results obtained by us on commercial horse spleen ferritin manufactured by Sigma-Aldrich Chemical Company.

### 2.5. Biological Research In Vitro

The objects of study were neutrophilic granulocytes of blood isolated from 29 apparently healthy blood donors and ferrihydrite nanoparticles, the concentrations of which were calculated based on the minimum and maximum amount of iron-containing substances in the human body. In this regard, the dose of nanoparticles in the minimum concentration was 25 mg, in the maximum—50 mg per 106 cells in 1 mL [30]. The donors’ health status and blood sampling were carried out based on the Blood Center No. 1 (protocol No 1, 13 January 2021). Before the blood sampling procedure, donors received a consent form to participate in research work.

Neutrophilic granulocytes were isolated from whole blood by layering on ficoll-urographin (ρ = 1.119) and centrifuged at 400 *g* for 45 min. The purity of the yield of neutrophilic granulocytes was determined by monitoring the morphological composition of leukocyte suspensions, which was 97%. The suspension of neutrophilic granulocytes was washed twice to remove plasma proteins in Hanks’ Balanced Salt solution without phenol red for 10 min at 400 *g*. The supernatant was discarded, the remaining neutrophilic granulocytes were diluted in 1 mL of Hanks’ solution and a suspension was obtained. The number of neutrophilic granulocytes in the hemocytometer was counted.

The functional activity of blood neutrophilic granulocytes was determined using chemiluminescent analysis. The study of spontaneous and zymosan-induced luminol-dependent chemiluminescence was carried out using a CL3606M biochemiluminescence analyzer (SKTB Nauka, Krasnoyarsk, Russia). The results of chemiluminescence analysis were characterized by the following parameters: the time to reach the maximum intensity (Tmax), the maximum value of the intensity (Imax), and the area (S) under the chemiluminescence curve. The enhancement of zymosan-induced chemiluminescence was assessed by the ratio of the induced area (Sind.) to the spontaneous area (Sspont.) and was defined as the activation index (AI).

Ferrihydrite nanoparticles were added into experimental samples immediately before chemiluminescence analysis at concentrations of 25 mg/mL and 50 mg/mL. The effect of ferrihydrite nanoparticles on cells after joint incubation for 30 min at 37 °C was evaluated. A test sample for determining a spontaneous reaction contained 200 μL of a suspension of neutrophilic granulocytes (the cell concentration was 106 per 1 mL), 20 μL of donor serum, 240 μL of Hanks solution, 50 μL of luminol. The experimental sample for determining the zymosan-induced reaction contained 200 μL of a suspension of neutrophilic granulocytes, 20 μL of donor serum, 200 μL of Hanks solution, 50 μL of luminol, and 40 μL of zymosan. As a control, the chemiluminescent activity of the spontaneous and zymosan-induced response was measured in neutrophilic granulocytes without being exposed to ferrihydrite nanoparticles.

Statistical analysis was carried out using the Statistica 6.1 software packages (StatSoft Inc., 2007, Tulsa, OK, USA). The description of the sample was performed by calculating the median (Me) and the inter-quarter range in the form of 25 and 75 percentiles (C25 and C75). The significance of differences between the indices of independent samples was assessed using the nonparametric Mann–Whitney test. The significance of differences between the indices of dependent samples was assessed by the Wilcoxon U test.

### 2.6. Biological Research In Vivo

The studies were conducted on sexually mature male Wistar rats weighing 200–250 g. The maintenance and manipulations of animals were carried out following the basic ethical principles in the field of bioethics [31].

Toxicological studies of the effect of ferrihydrite nanoparticles on tissues and organs of laboratory animals were carried out. The concentration of ferrihydrite nanoparticles in the preparation was 2.4 g/L. The dose of drug administration was 0.2 mL per animal every 8 days for 1 month. Ferrihydrite nanoparticles in the form of an aqueous suspension were injected intramuscularly into the thigh.

Following the objectives of the study, laboratory animals were divided into 2 groups (*n* = 8 each): (1) animals without exposure; (2) animals that were injected with an aqueous sol of nanoparticles of ferrihydrite.

At the end of the experiment, all animals were euthanized. The material for histological examination was the samples of the liver, spleen, and kidneys. Histological preparations were prepared from samples and stained with hematoxylin and eosin for review purposes and determination of iron (III) compounds using Perls Prussian blue method [32,33].

## 3. Results

### 3.1. Dynamic Light Scattering

Measurements of dynamic light scattering were performed on a Zetasizer Nano apparatus (Malvern Instruments Ltd., Malvern, United Kingdom, HeNe laser, λ = 632.8 nm, Krasnoyarsk Regional Center of Research Equipment of Federal Research Center “Krasnoyarsk Science Center SB RAS”). The distribution of nanoparticles obtained by dynamic light scattering (Figure 2) has a polymodal form with modal values of 28.2, 105, 7, and 220.2 nm. The size of ferrihydrite nanoparticles determined using a transmission electron microscope is 2–7 nm. Nanoparticles registered by dynamic scattering are aggregates. The fraction of aggregates of nanoparticles with a hydrodynamic diameter of ~28 nm is 65.4%, and the fraction of large nanoparticles is 34.6%. The large zeta potential (82 mV) indicates that the sol of ferrohydrite nanoparticles is highly stable.

### 3.2. FTIR Spectroscopy

The IR spectrum of dried sol of ferrihydrite nanoparticles produced by *Klebsiella oxytoca* bacteria is presented in Figure 3.

IR spectrum of ferrihydrite nanoparticles is featured by the broad absorption band maximizing at 3420 cm^−1^ that is assigned to Oh group within the crystal structure of ferrihydrite. The large width of this band is partially due to surface O–H bonds of adsorbed water molecules. Bending vibrations of H_2_O are pronounced at 1630 cm^−1^ [34]. Our sample contains organic molecules, and the spectral range 1300–1700 cm^−1^ is featured by two broad absorption bands peaking at 1498 and 1410 cm^−1^ assigned to carboxyl groups. The spectral range 900–1200 cm^−1^ is known as the “fingerprint” range. The spectrum of the sol under investigation contains the most intense bands in this range, that are the manifestations of stretching vibrations of C–C groups as well as symmetric and asymmetric vibrations of valence bonds C–O. In our sample, the vibrations of Fe-O bonds fall into this spectral range [34]. Weak absorption at 874 cm^−1^ can be an indication of the presence of glucose. Vibrations at 564 and 604 cm^−1^ are assigned to TO–LO modes associated with α-Fe_2_O_3_ [35].

### 3.3. Mössbauer Spectroscopy

Figure 4 shows the Mössbauer spectra and probability distributions of quadrupole splitting P(QS) of samples, depending on the duration of cultivation on media containing various forms of iron (Fe^2+^ oxalate and Fe^3+^ citrate). 

The maxima and features on the P(QS) distributions indicate possible nonequivalent positions of iron. Based on the P(QS) shape, the sum of three doublets with different quadrupole splittings is taken as the model spectrum. The model spectrum was fitted to the experimental spectrum by varying the entire set of hyperfine parameters. As a result of such decoding of the spectra, three types of ferric iron positions are identified: Fe1 with QS ~0.55 mm/s, Fe2 with QS ~1 mm/s, and Fe3 with QS ~1.5–1.8 mm/s. The origin of these positions is explained as follows. The crystal–chemical model of ferrihydrite is a trigonal layered structure with dense packing of anions [36] with a large number of stacking faults. In this case, fragments with cubic ABCABC and hexagonal ABAB (or ACAC) packing of anion layers are formed. Upon cubic packing of anions, two adjacent layers of octahedra are formed, occupied by iron cations Fe^3+^. Here, the anionic octahedra are connected by edges, both in the layer and between the layers, providing a relatively weak distortion of the octahedra associated with the Fe1 position. When a fragment with hexagonal packing is formed, single layers of iron-occupied octahedra are formed [37], which are interconnected by edges only inside the two-dimensional layer, forming distorted Fe2 positions. When iron cations exit from occupied layers to empty ones, the pairs of octahedra bound by faces are formed. Due to the small cation–cation distance, the iron cations tend to push off from each other, displacing from the centers of the octahedra, thereby providing a large quadrupole splitting for the Fe3 sites. Moreover, these cations have a random environment in the second neighbors. Figure 5 shows the dependences of the occupancies of the detected sites, isomeric shifts (IS), and quadrupole splittings (QS) on the duration of cultivation.

### 3.4. Magnetic Properties of Ferrihydrite Nanoparticles

Figure 6 shows the temperature dependences of the magnetization measured in the ZFC and FC modes for dry sol of ferrihydrite nanoparticles and with commercial ferritin taken as the reference. These dependences demonstrate the features characteristic for systems of superparamagnetic (SPM) particles: a clearly expressed maximum of the M(T)_ZFC_ dependence, and bifurcation of the dependences M(T)_ZFC_ and M(T)_FC_ which begins in the vicinity of T_max_. For ferritin, the T_max_ value is ≈13 K, for the ferrihydrite nanoparticles, ≈23 K. The temperature T_max_ can be logically considered as the temperature of the SPM blocking. At this temperature, the system of magnetic moments of nanoparticles passes from a blocked (at T < T_max_) to an unblocked (at T > T_max_) state. According to our data, for the nanoparticles of ferrihydrite, the temperature T_max_ for samples of different batches (prepared at different times) varied by no more than 3 K [9,38,39].

Figure 7 shows the magnetization curves M(H) for ferrihydrite nanoparticles and ferritin. At temperatures below T_max_, the dependences M(H) exhibit hysteresis. The coercive force H_C_ at T = 4.2 K for ferrihydrite nanoparticles is ≈2.1 kOe, which exceeds H_C_ for ferritin (see inset in Figure 7). For T > T_max_, the dependences M(H) are completely reversible. Their shape is described by the Langevin function corresponding to SPM behavior, and the contribution is linear on the field, corresponding to the antiferromagnetically ordered “core” of particles [9,38,39,40].

### 3.5. Investigation of the Effect of Ferrihydrite Nanoparticles In Vitro

Luminol-dependent chemiluminescence was used to determine the basic activity of cells and the reserve capacity of neutrophilic granulocytes when exposed to a nonspecific inductor in the form of zymosan. The ability of neutrophilic granulocytes to form a general pool of secondary oxygen radicals (H_2_O_2_, OH, O_2_, HClO) was studied. The study of the lucigenin-dependent reaction determined the activity of the superoxide anion radical [30].

The study of the chemiluminescent response of neutrophilic granulocytes to the effect of nanoparticles at a minimum concentration revealed only a change in the lucigenin-dependent chemiluminescence response relative to the control group. An increase in the intensity of the process and the activation index in the zymosan-induced process (by four and three times, respectively) was revealed. Thus, the functional activity of neutrophilic granulocytes changes only concerning the superoxide anion radical when exposed to ferrihydrite nanoparticles at a minimum concentration (Table 1).

Assessment of the activity of neutrophilic granulocytes on the effect of the maximum concentration of ferrihydrite nanoparticles revealed a statistically significant decrease (½ times) in the indicators of maximum intensity in the zymosan-induced process. The area under the curve of the chemiluminescent response stimulated by zymosan also decreased by two times relative to the control. In the case of zymosan-induced chemiluminescence, the index of activation (IA) was decreased twice when exposed to nanoparticles (Table 2).

The study of the reaction of neutrophilic granulocytes to the effect of ferrihydrite nanoparticles in the minimum concentration after joint incubation for 30 min also revealed an increase in the intensity and area under the curve of the lucigenin-dependent chemiluminescent process relative to the control (by three and five times). In the case of exposure to the maximum concentration of nanoparticles, a decrease in the indicator characterizing the time to peak (10 times) in the spontaneous chemiluminescence reaction and in the indicator of the maximum intensity of the chemiluminescence process (six times) was found. The chemiluminescence index, which characterizes the maximum area under the curve of the zymosan-induced reaction, decreased by five times after the incubation of neutrophilic granulocytes with nanoparticles. Long-term exposure to ferrihydrite nanoparticles resulted in a three-fold decrease in IA.

### 3.6. Investigation of the Effect of Ferrihydrite Nanoparticles In Vivo

The intramuscular injection of aqueous sol of ferrihydrite did not cause clinically pronounced negative changes. Microscopic examination showed that in all samples of organs and tissues of the experimental group, similar changes are observed, which are a consequence of the deposition of iron upon its excessive intake.

#### 3.6.1. Morphological Changes in the Liver

The study of liver samples of the control group corresponds to the normal structural organization of the organ. The liver is of a typical structure: the beam structure is preserved, the central veins, sinusoidal capillaries are unchanged. Most hepatocytes are mononuclear, a small number of hepatocytes are also found. The portal tracts are not thickened. During the Perls reaction in the liver, the formation of Prussian blue was not observed.

In animals treated with a sol of ferrihydrite nanoparticles, histological changes in liver tissue were detected. Despite a slight expansion of the central veins and sinusoidal capillaries filled with protein light eosinophilic mass, hepatocytes were in a state of moderate protein dystrophy (Figure 8) and an increase in the volume of cells with two nuclei was noted, which may indicate the beginning of structural compensation of damaged tissue by cell proliferation. Kupffer cells are large, irregular in shape, their number is increased. Kupffer cells are found throughout the sinusoidal capillaries and around the portal tracts. The Perls reaction revealed a partial distribution in the parenchyma of the organ of inclusion of crystals of Prussian blue (Figure 9), which indicates that there is no accumulation of nanoparticles of ferrihydrite in the organ and their rapid excretion from the organ.

#### 3.6.2. Morphological Changes in the Spleen

The study of histological sections of the spleen from rats of the control group shows a typical structure. The organ parenchyma is represented by clearly distinguishable red and white pulp. White pulp is characterized by the presence of formed lymphatic follicles with pronounced germinal centers, periarterial, mantle and marginal zones. Histological examination of tissues of intact animals showed that in the spleen during the Perls reaction formation of Prussian blue was not observed.

In the spleen of rats receiving a sol of ferrihydrite nanoparticles, the distribution of nanoparticles was noted both in the parenchyma and in the organ stroma (Figure 10). The deposition of nanoparticles of ferrihydrite is observed both in red and partially in white pulp. In the spleen, an increase in the area of red pulp is noticed with respect to white pulp. Plethora was pronounced in the red pulp. The follicles had different sizes; in the individual follicles, light centers and a fuzzy mantle zone were found. The endothelium of the central arteries of the follicles is swollen, the vessel wall is loose, the arteries themselves are partially in a state of hyperemia. Damage to the vascular endothelium was systemic, as was observed in most cases.

#### 3.6.3. Morphological Changes in the Kidneys

In the animals of the control group, a normal histological picture of the structure of the renal cortex and medulla was observed. The vascular glomeruli are not enlarged, the basement membrane is thin, the space between the capsule and glomerulus is preserved. The openings of the straight and convoluted tubules are not enlarged, the epithelial cells of the proximal convoluted tubules are cubic, pink cytoplasm, spherical cell nuclei. Histological examination of tissues of intact animals showed that during the Perls reaction the formation of Prussian blue in the kidney was not observed.

In the kidneys of the rats endured the introduction of an aqueous sol of ferrihydrite nanoparticles (Figure 11), moderate glomerular congestion of the glomeruli, and swelling of the convoluted tubule epithelium was observed in areas where partial deposition of Perl-positive granules was revealed.

#### 3.6.4. Morphological Changes in the Lungs

In the control group of animals at the end of the experiment, the histological examination did not reveal significant violations in the structural organization of the lung. The histological structure of the organ of animals treated with a suspension of ferrihydrite nanoparticles was unchanged. The cavities of most of the alveoli are straightened, optically empty. The interalveolar walls are not thickened, their capillaries are moderately congested, and paretic congestion of larger vessels is noted. There are scattered small areas in which the alveolar cavities are widened and optically empty, the interalveolar walls are thinned and partially torn. These are foci of acute emphysema. The lumens of the small bronchi are dilated, free, with a moderately pronounced focal peribronchial lymphoid infiltration around (Figure 12).

Alveolar macrophages are detected partially perivascular and peribronchial and are also noted in interalveolar septa. They are represented by a few cells containing a small number of ferrihydrite nanoparticles, which give a positive Perls reaction.

## 4. Discussion

The dependences of the dry weight of bacteria cultured under anaerobic conditions on media containing various forms of iron (Fe^2+^ oxalate and Fe^3+^ citrate) presented in Figure 1 clearly demonstrate that 5 days for cultivation on iron citrate are sufficient for the practical production of bacterial sediments. Corresponding indicators of dry weight during cultivation on oxalate are achieved only after two weeks of cultivation.

The results of Mössbauer spectroscopy correlate with the results of the dry weight production of bacterial sediments. At the initial stages of cultivation (3–14 days), the values of isomeric shifts (IS) and the nature of the changes in Mössbauer spectra of precipitates formed during the cultivation of microorganisms on Fe^2+^ oxalate and Fe^3+^ citrate differ significantly. When cultured on oxalate, the IS of Fe^2+^ for three detected sites is 0.39 mm/s for 3–6 days. When cultured on citrate, the Fe^3+^ IS is about 0.41 mm/s. Only starting from 14 days do the parameters of IS precipitation during cultivation on Fe^2+^ oxalate (0.42 mm/s) reach the parameters of IS precipitation during cultivation on Fe^3+^ citrate after 3–6 days. With a further increase in the cultivation duration, the values of isomeric shifts monotonously decrease. Thus, it is seen from the analysis of IS dependences that the process of production of nanoparticles of ferrihydrite by microorganisms during cultivation on Fe^2+^ oxalate is delayed by 14–15 days compared to cultivation on Fe^3+^ citrate. The dependences of the quadrupole splitting QS and the occupancies of the detected positions of Fe(1), Fe(2), Fe (3) for the discussed cultivation modes correlate with each other.

The organic matrix of the studied sol of biogenic nanoparticles *Klebsiella oxytoca* deserves particular attention for practical applications. In [41], where the dried biomass of *Klebsiella oxytoca* bacteria was studied by IR spectroscopy, it was suggested that there were glucose, polysaccharides, and amines in the sample. The IR spectra of proteins and their decay products (peptides) feature by the presence of two main absorption bands, namely amide I at 1650 cm^−1^ and amide II at 1550 cm^−1^, due to stretching vibrations of the C = O bond (amide I) and planar bending vibrations of the NH bond (amide II) [42]. Thus, the manifestation of the characteristic frequencies in the IR spectrum at 1650 cm^−1^ and 1550 cm^−1^ makes it possible to detect the presence of amides in the analyzed sample. In the spectrum of the sol studied by us, modes characteristic of amides are not observed, but absorption bands characteristic of polysaccharide structures are present.

Taking into account the distribution of magnetic moments in magnitude allows one to rather accurately estimate the value of the average uncompensated moment μ_P_ of ferrihydrite and ferritin particles [6,9,38,39,43,44] from experimental magnetization curves in the region T > Tmax. For this, the dependences M(H) (Figure 7) were fitted by the function M(H) = NP ∫ L(μ_P_) f(μ_P_) + C-H (TrueType), in which NP is the number of particles per unit mass of the sample, L(μ) is the Langevin function, f(μ_P_) is the distribution function of magnetic moments (the lognormal distribution was used), and the term C-H is responsible for the antiferromagnetic canting of the sub lattices and other contributions [40,45,46,47,48]. For dried sol, the μ_P_ value varies in the range 160–230 μ_B_ (μ_B_ is the Bohr magneton) for samples of different batches [9,38,39]. The obtained μ_P_ values are explained in the framework of the Néel hypothesis [49] for small antiferromagnetic particles, according to which μ_P_~μ_at_⋅N^b^, where μ_at_ is the magnetic moment of the atom, N is the number of magnetically active atoms in the particle, and the exponent b is 1/3, 1/2, and 2/3 in the case of defects on the surface, in the volume of the particle, and an odd number of ferromagnetic ordered planes, respectively. For the obtained ferrihydrite, N ~ (1.5–2) × 10^3^, μ_Fe_ ≈ 5 μ_B_. From these data, we obtain agreement between the obtained μ_P_ (160–230 μ_B_) and the Néel model hypothesis for b ≈ 1/2, which corresponds to the presence of defects in the particle volume. A similar conclusion was obtained in the analysis of data on ferritin [6,50].

An in vitro study of the effect of ferrihydrite nanoparticles on the functional activity of neutrophilic granulocytes has shown that at the first stage of cell response, the antioxidant potential of cells is activated, which contributes to their protection from reactive oxygen species. Consequently, the first reaction to nanoparticles in cells is the activation of a defense mechanism aimed at reducing the concentration of free radicals. This is indicated by the low response of zymosan-induced chemiluminescence of cells after incubation with nanoparticles at the maximum concentration, and this reflects the low activity of oxygen radicals and thus a decrease in the level of cell metabolism upon prolonged exposure to nanoparticles. It is also interesting that the minimum concentrations of ferrihydrite nanoparticles increase the activity of primary oxygen radicals, while the maximum concentrations decrease the production of secondary radicals.

Thus, the effect of nanoparticles on phagocytes significantly reduces the level of free radical oxidation. The short-term effect of nanoparticles can be modulating and depend on the initial level of cell reactivity. A unique feature is that the reduction in oxidative stress due to nanoparticle exposure occurs exclusively in a stimulated response. Thus, the effect of nanoparticles is realized mainly on activated cells.

The results of testing the biogenic water sol of ferrihydrite on laboratory animals indicate that the toxicokinetics (retention, redistribution, and elimination) of nanoparticles are controlled by both physiological and physicochemical mechanisms, namely active endocytosis carried out by various cells and gradual excretion from the body.

Thus, the effect of biogenic ferrihydrite nanoparticles can be explained, on the one hand, by the ability to penetrate through biological barriers (dispersion of nanoparticles), the ability of nanoparticles to directly penetrate from the sites of primary deposition into the bloodstream with subsequent retention in internal organs, especially in organs rich in cells of the reticuloendothelial system, particularly in the spleen and partly in the liver. On the other hand, the effect of nanoparticles is explained by the ability of iron nanoparticles to release toxic ions near ultrastructural and molecular targets for their impact (damage endothelium of the vessels of the spleen, and in the liver and kidneys of parenchyma cells with the development of mild and moderate granular dystrophy in them). Perhaps these processes are the result of a free radical oxidation due to the presence of an increased amount of iron, which has an influence on the formation of free radicals. As a consequence, increased oxidation can stimulate inflammatory processes in the organs and tissues of the body as a result of the release of iron ions.

Along with the negative effect of the sol nanoparticles on organs and tissues, one should note their positive effect, associated with the activation of compensatory–adaptive processes in the studied organs and tissues, characterized by an increase in the number of binuclear hepatocytes, Kupffer cells, as well as increased hematopoiesis, which is manifested by an increase in extramedullary foci of hematopoiesis in the liver and spleen.

Elimination of biogenic ferrihydrite nanoparticles consists of three main mechanisms: renal, hepatobiliary, and through the mononuclear phagocytic system. In the liver, the elimination of nanoparticles is carried out by endocytosis, followed by their enzymatic cleavage and excretion into bile through the biliary system. The accumulation of Kupffer cells in the liver is periportal, i.e., migration occurs from the sinusoids to the biliary tract of the portal tracts, followed by penetration into their lumen. These observations are consistent with the literature data on the existence of a mechanism for the elimination of nanoparticles from the body in bile [51].

The spleen has the largest accumulation of ferrihydrite nanoparticles compared to other organs. This indicates the active participation of the spleen in the processes of elimination of nanoparticles from the blood and their accumulation in the cells of the mononuclear phagocyte system (monocytic–macrophage system). No significant morphological changes were found in the spleen. This indicates the broad compensatory capabilities of this organ. The spleen protects other organs from the effects of nanoparticles by decreasing their concentration in the blood.

In the kidneys, discirculatory changes are weakly expressed, which indicates the implementation of adaptive mechanisms. The relatively small number of Perls-positive cells indicates that the kidneys do not accumulate ferrihydrite nanoparticles.

In the rat lung, macrophages that give a positive Perls reaction lie perivascular, which indicates the interaction of these cells with ferrihydrite nanoparticles immediately after their penetration into the tissue from the lumen of the vessel. Single accumulations of Perls-positive macrophages are found in the interalveolar septa and peribronchial, which indicates the interaction of alveolar macrophages with ferrihydrite nanoparticles, namely migration into the lumen of the bronchial tree. A small number of Perls-positive macrophages indicates the elimination of nanoparticles by phagocytosis and their removal from the lung by the transport system.

## 5. Conclusions

In this work, nanoparticles of biogenic ferrihydrite synthesized as a result of the cultivation of *Klebsiella oxytoca* bacteria were studied. Using the method of Mössbauer spectroscopy, the optimal conditions and duration of the synthesis were determined. The study of the organic shell covering the surface of the nanoparticles by IR spectroscopy showed that it consists of polysaccharides. According to the static magnetic properties, ferrihydrite nanoparticles are close to a commercial product—horse spleen ferritin. The uncompensated magnetic moment of the ferrihydrite particles amounts to about 200 Bohr magnetons, which corresponds to ~ 40 uncompensated spins in a particle with N ~ 1.5–2 × 10^3^ iron atoms. This is consistent with the Néel hypothesis, in which the presence of defects in the volume of a small antiferromagnetic particle leads to the appearance of its magnetic moment proportional to N^1/2^ [49]. At room temperature, the ferrihydrite nanoparticles are superparamagnetic, and the characteristic temperature of the SPM blocking is 23–25 K.

In vitro testing has shown a decrease in the functional activity of blood phagocytes when exposed to ferrihydrite nanoparticles that did not contact plasma proteins. In vivo testing of ferrihydrite nanoparticles was shown that the water sol has chronic toxicity since it leads to morphological changes in the organs, mainly the spleen, which is characterized by the accumulation of iron nanoparticles in the form of hemosiderin, which gives a positive Perls reaction [52].

The positive effect of nanoparticles of ferrihydrite is reflected in the activation of phagocytosis in the liver. According to our data, iron nanoparticles have another important property, directly or indirectly, to stimulate the proliferation and intracellular regeneration of hepatocytes. The partial detection of Perls-positive cells in the liver and kidneys can be explained by the rapid excretion from organs and the high dispersion of the nanomaterial.

Thus, the introduction of nanoparticles into the body is characterized by adaptive-proliferative processes, accompanied by the development of cell dystrophy and tension of the phagocytic system. The biological effects we observed may be associated with the formation of a protein corona on the surface of nanoparticles; however, further studies are required for a more complete and deeper understanding.

Nanoparticles can accumulate and remain in the body for a fairly long time, and their interaction with biological objects is largely determined by the properties of nanoparticles. Therefore, it is necessary to investigate these processes at the systemic level and involve methods for predicting the properties of nanoparticles and modeling the mechanisms of interaction of nanoparticles with biological molecules and systems.

## Figures and Tables

**Figure 1 biomedicines-09-00323-f001:**
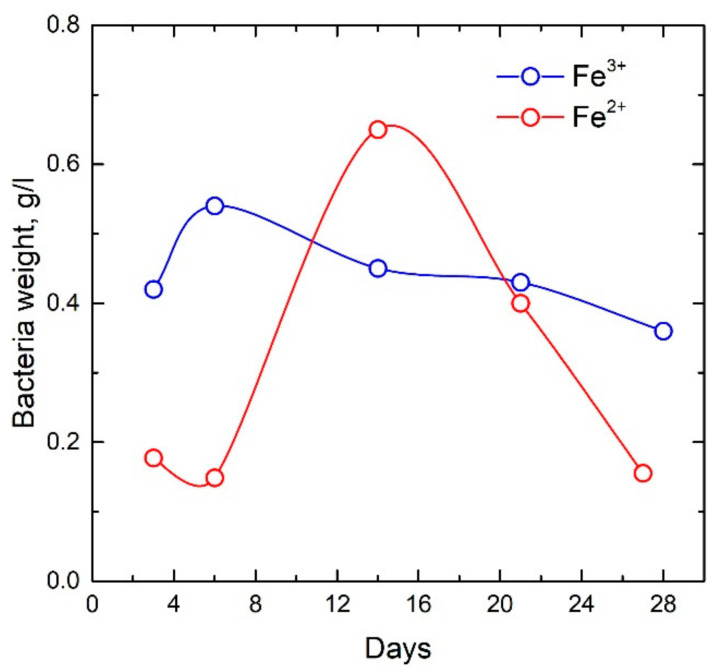
Time dependences of the dry weight of bacteria grown on Fe^2+^ oxalate and Fe^3+^ citrate.

**Figure 2 biomedicines-09-00323-f002:**
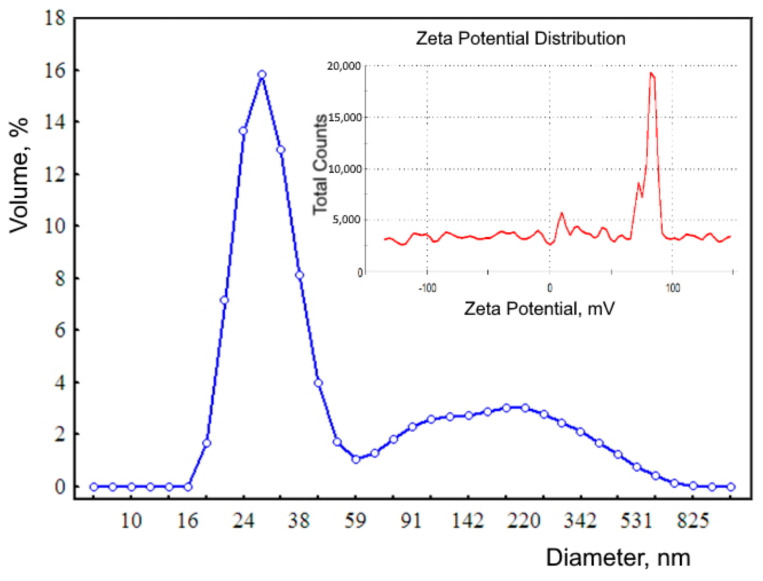
Distribution of the hydrodynamic diameter of biogenic ferrihydrite (blue line). Inset: zeta potential distribution (red line).

**Figure 3 biomedicines-09-00323-f003:**
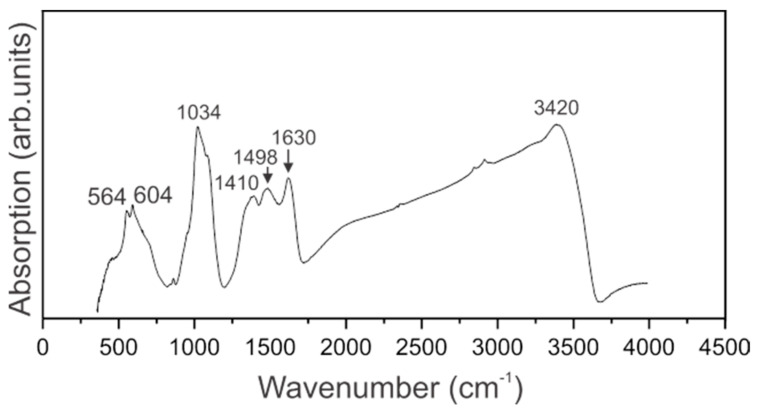
IR-Fourier spectrum of a sol of ferrihydrite nanoparticles.

**Figure 4 biomedicines-09-00323-f004:**
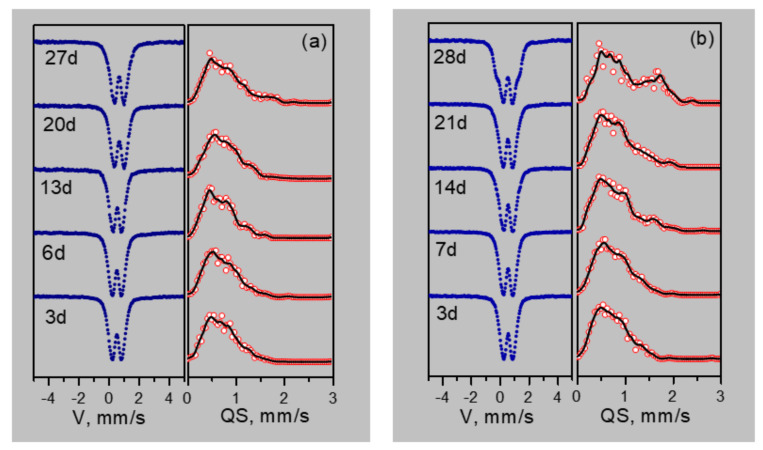
Mössbauer spectra and distributions of quadrupole splitting of nanoparticles depending on the duration of cultivation on media containing various forms of iron (Fe^2+^ oxalate (**a**) and Fe^3+^ citrate (**b**)).

**Figure 5 biomedicines-09-00323-f005:**
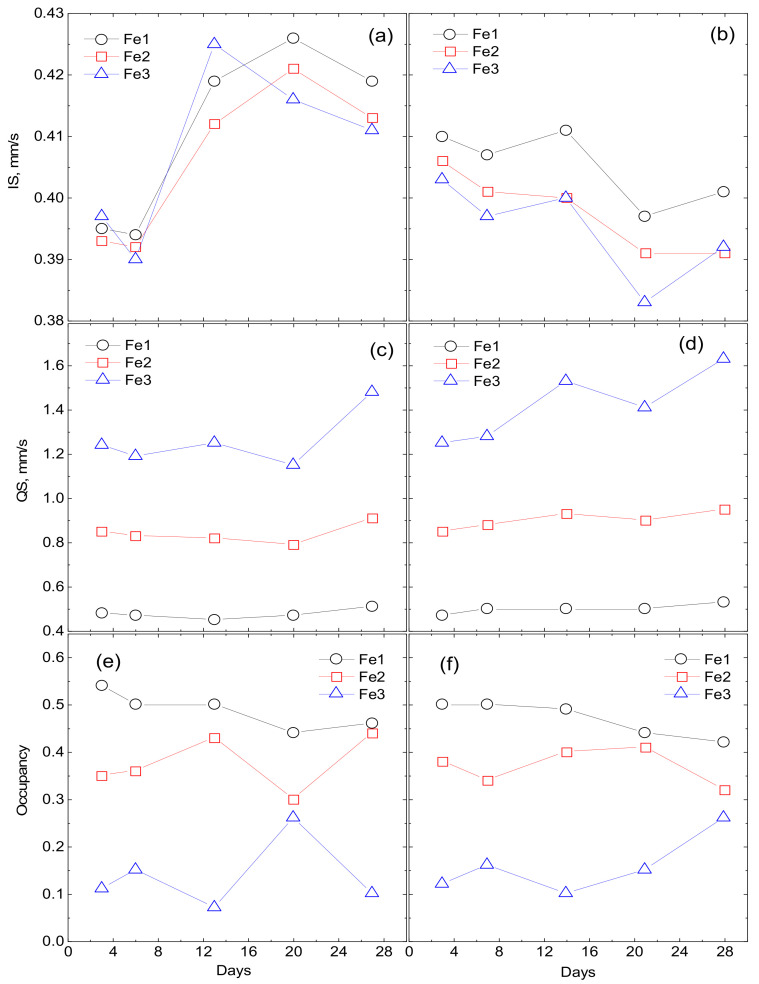
Dependencies of site occupancies, isomeric shifts (IS) and quadrupole splittings (QS) on the cultivation duration. Fe^2+^ denotes oxalate medium (**a**,**c**,**e**) and Fe^3+^ denotes citrate medium (**b**,**d**,**f**).

**Figure 6 biomedicines-09-00323-f006:**
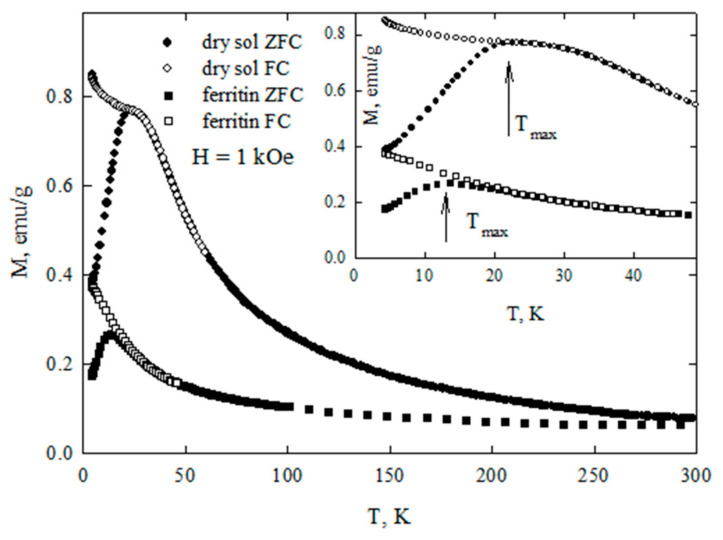
Dependencies of magnetization M(T) of dry sol of ferrihydrite nanoparticles and of commercial ferritine under zero-field cooling (ZFC) and field cooling (FC) conditions. Inset: M(T) in the temperature range of superparamagnetic (SPM) blocking.

**Figure 7 biomedicines-09-00323-f007:**
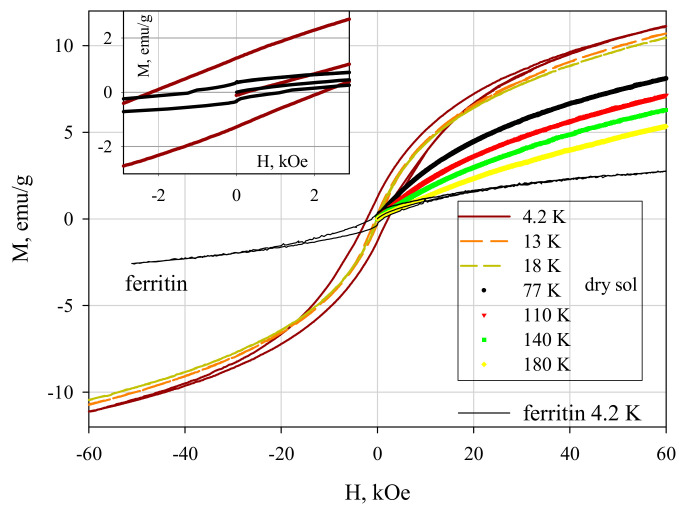
Dependence of magnetization M(H) of dried sol of ferrihydrite nanoparticles and of commercial ferritine at different temperatures. Inset: M(H) in the vicinity of H = 0.

**Figure 8 biomedicines-09-00323-f008:**
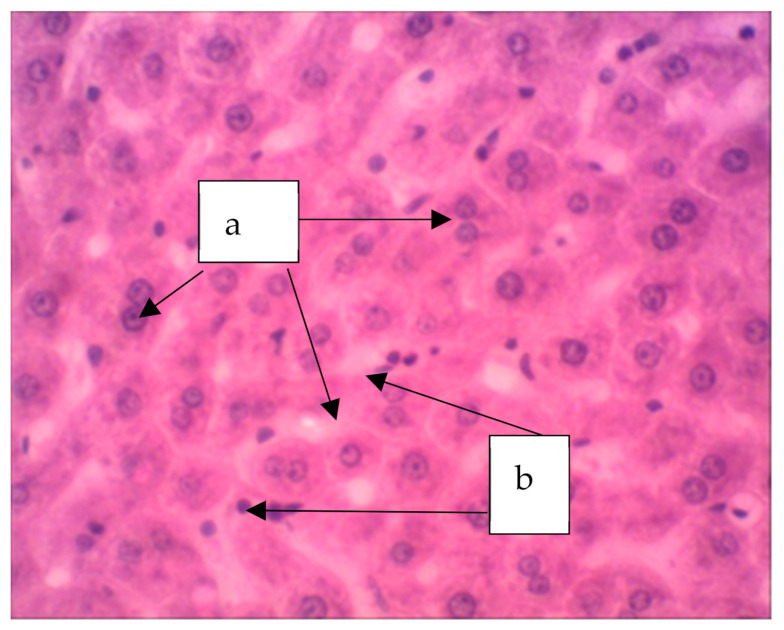
The liver of a rat. Increase in the volume of hepatocytes with two nuclei (a), increase in volume of Kupffer cells (b). Staining with hematoxylin–eosin, ×400.

**Figure 9 biomedicines-09-00323-f009:**
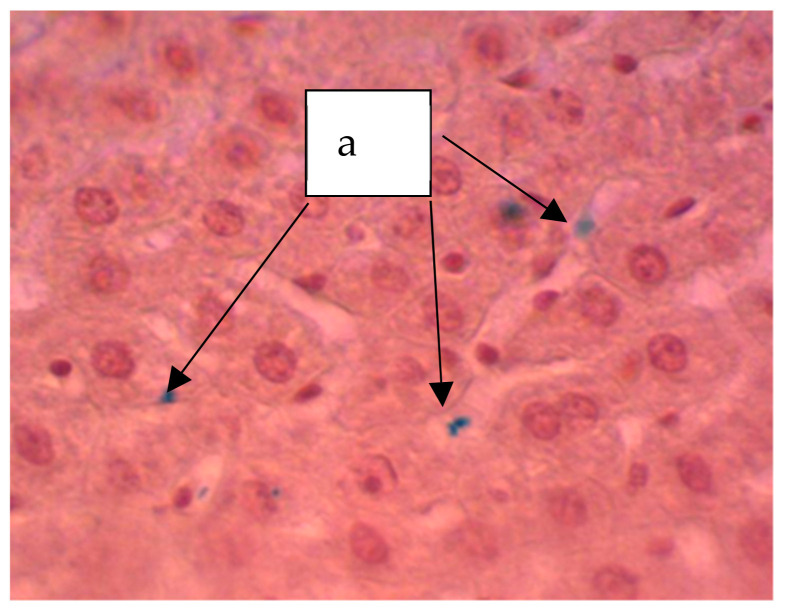
Liver of a rat. Staining according to the Perls method. The particles stained blue can be seen (a). ×400.

**Figure 10 biomedicines-09-00323-f010:**
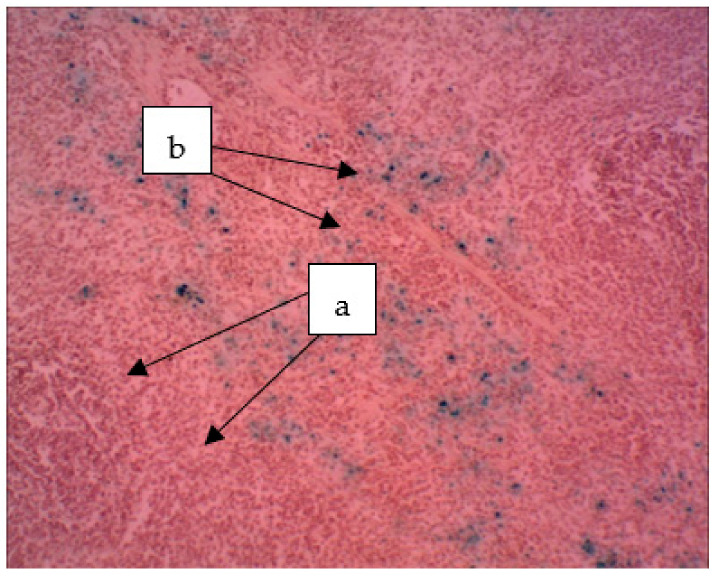
The spleen of a rat. The deposition of hemosiderin grains in the red pulp (a) and hemosiderin in the white pulp (b). The particles stained blue can be seen (a). ×400.

**Figure 11 biomedicines-09-00323-f011:**
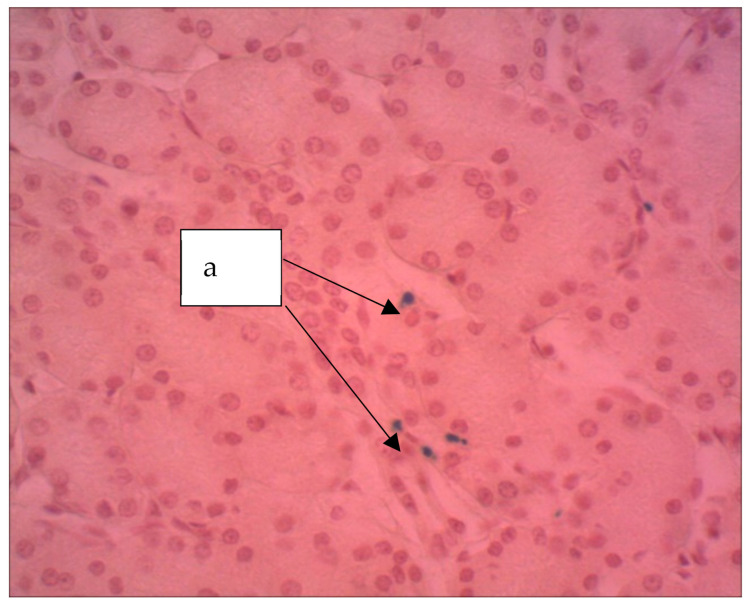
Kidney of a rat, Hemosiderin grains are seen (a). The particles stained blue can be seen (a). ×400. Staining with hematoxylin–eosin.

**Figure 12 biomedicines-09-00323-f012:**
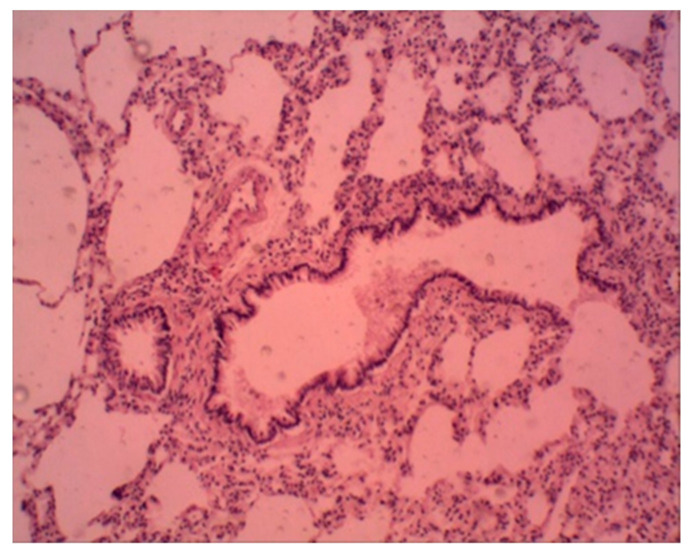
Rat lung. Foci of emphysema, peribronchial lymphoid infiltration. Staining with hematoxylin–eosin. ×100.

**Table 1 biomedicines-09-00323-t001:** Indicators of luminol-and lucigenin-dependent chemiluminescence of neutrophil granulocytes under the influence of different concentrations of ferrihydrite nanoparticles without incubation.

Characteristics ^1^	Control	Ferrihydrite Nanoparticles (25 mg/mL)	Ferrihydrite Nanoparticles (50 mg/mL)
	1	2	3
Luminol-dependent reaction			
I_max_	20,060 (12,451–32,490)	3321 (988–7673)	11,384 (2807–19,278) P_1_ = 0.014
S_max_ × 10^5^	7.1 (3.8–10.4)	1.2 (0.3–2.7)	3.4 (1.02–5.9) P_1_ = 0.006
IA	3.1 (2.2–4.3)	5.7 (1.4–9.5)	9.3 (6.7–11.5) P_1_ = 0.009
Lucigenin-dependent reaction			
I_max_	2009 (919–2640)	8170 (7580–9030) P_1_ < 0.001	183 (886–3044)
IA	2.3 (1.1–3.4)	6.5 (4.3–7.0) P_1_ < 0.001	3.1 (2.4–4.3)

^1^ I_max_, maximum intensity; S_max_, maximum area; IA, index of activation.

**Table 2 biomedicines-09-00323-t002:** Indicators of luminol-and lucigenin-dependent chemiluminescence of neutrophil granulocytes under the influence of different concentrations of ferrihydrite nanoparticles with incubation.

Characteristics ^1^	Control	Ferrihydrite Nanoparticles (25 mg/mL)	Ferrihydrite Nanoparticles (50 mg/mL)
	1	2	3
Luminol-dependent reaction			
Spontaneous reaction			
T_max_	7680 (5580–127,300)	8230 (5040–10,560)	791 (496–1992) P_1_ < 0.001
Zymosan-induced reaction			
I_max_	32,061 (15,452–34,571)	14,234 (4917–16,348)	5454 (1071–6684) P_1_ = 0.014
Smax × 10^5^	9.3 (5.7–10.8)	6.4 (4.8–8.9)	1.7 (0.5–2.4) P_1_ < 0.001
IA	1.0 (0.6–1.2)	1.3 (1.0–2.5)	3.1 (2.3–4.1) P_1_ < 0.001
Lucigenin-dependent reaction			
I_max_	1619 (1219–2568)	4860 (3571–7123) P_1_ < 0.001	1454 (796–2064)
S_max_ × 10^4^	2.8 (1.9–3.3)	10.8 (9.1–12.1) P_1_ < 0.001	3.0 (2.3–3.3)

^1^ T_max_, time of maximum activity; I_max_, maximum intensity; S_max_, maximum area; IA, index of activation.

## Data Availability

The data presented in this study are available on request from the corresponding author.

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
