# Peer review of "Biogenic Ferrihydrite Nanoparticles: Synthesis, Properties In Vitro and In Vivo Testing and the Concentration Effect"

_biomedicines, 2021, doi:10.3390/biomedicines9030323_

Round 1

Reviewer 1 Report

Review report on the manuscript ID :biomedicines -1118289

Titled: Biogenic Ferrihydrite Nanoparticles: Synthesis, Properties in Vitro and in Vivo Testing and the Concentration Effect

The paper describes the production of biogenic ferrihydrite nanoparticles as a result of the cultivation of Klebsiella oxytoca microorganisms. FTIR Spectroscopy, Mossbauer Spectroscopy, Magnetometric Measurements were applied in order to characterize the nanoparticls from the physico-chemical point of view. In vitro testing of different concentrations of ferrihydrite noparticles were performed for the functional activity of neutrophilic granulocytes by the chemiluminescence  method, while In vivo testing  on Wister rats showed  chronic toxicity and morphological changes in organs, mainly in the spleen.

General comments: I consider that the subject of the manuscript is more suitable for another jurnal in MDPI, which is specifically devoted to the production, characterization and biomedical application of nanoparticles. For example, the Special Issue “Preparation, physico-chemical Properties, and Biomedical Applications of nanoparticles” in MATERIALS, MDPI-Journal.

Specific comments:

  1. The characterization of prepared nanoparticles is not complete without DLS measurement, including also Zeta potential determination.
  2. As there are many papers dealing with the preparation and different medical applications of Fe2O3 nanoparticles, the authors should clearly point out the novelty of their work, if there is any.
  3. The toxicity of Fe2O3 particles is wellknown. Maybe the authors will consider to explain a new mechanism of action? Otherwise, what is the relevance of the study?

In conclusion, I recommend the re- submission of the manuscript to another journal, more suitable with respect to the subject.

Author Response

Уважаемый рецензент, большое вам спасибо за столь подробное рассмотрение статьи, замечания и предложения, которые помогли улучшить качество статьи.

Reviewer 2 Report

The authors synthesized the ferrihydrite nanoparticles and studied its propeties in vitro and vivo. The manuscript could be considered to be accepted after minor revision for some typos: the superscripts and subscripts for the unit (cm2), chemical formulas (Fe2+, Fe3+), ... The figure should be clearer (Figure 1).

Author Response

Dear reviewer, we are grateful for a thorough study of the paper and for comments and suggestions that helped us improve the quality of the paper. We took into account your comments and made changes to the paper.

Reviewer 3 Report

In this manuscript authors present biogenic synthesis of ferrihydrite nanoparticles and their physicochemical and biological evaluation.

The manuscript presents a very interesting topic of research. It is very appreciated that authors showed and discussed honestly the results not focusing only on “positive” outcomes, but highlighting potential toxicological problems. In a light of avalanche of “positive” only reports, this particular study requires attention.

However, major changes are needed before it is ready for publication.

Firstly, authors should add more detailed information on nanoparticle characterization, e.g. zeta-potential, EDX analysis, stability in biological liquids and size change upon interacting with plasma proteins. Please follow the guidelines [1].

At present the Introduction, Main text and Discussion are not as scholarly and sceptical in its presentation of previous work as it should be. Discussion needs extensive revision. It is important to explain the rationale with merging with the literature. Therefore, authors are urged to discuss following current literature about nanoparticles.

Authors should comment in detail about protein corona. It becomes evident in nano-bio field that nanomaterials at fist interact with proteins of biological liquids and such interactions change completely desired biological behaviour of nanomaterials. There are numerous studies clearly showing, that when nanoparticles are introduced into biological systems, the surface of nanoparticle immediately interacts with proteins and other biomolecules forming so-called protein corona [2, 3]. The formation of protein corona has important consequences on nanoparticle hydrodynamic diameter, solubility, protein misfolding and aggregation [2-5]. However more importantly, corona also masks the desired biological functionalities [3]. It has been shown that protein corona caused loss of the intended targeting properties of the functionalized nanoparticles [6] and enzymes bound to the surface of nanoparticles change their activity [7, 8]. As a result, it is very hard to make any reasonable conclusion taking into account “naked” nanoparticles.

Overall problem of liver-nanoparticle interaction, excretion or not well discussed. What type of cells are involved, what are the routes for internalization excretion, etc? For the examples see [9-11]. I understand that this is not the major topic of the paper, however at least briefly it should be discussed.

  1. Faria, M., et al., Minimum information reporting in bio-nano experimental literature. Nat Nanotechnol, 2018. 13: p. 777-785.
  2. Del Pino, P., et al., Protein corona formation around nanoparticles - from the past to the future. Materials Horizons, 2014. 1: p. 301-313.
  3. Ke, P.C., et al., A decade of the protein corona. ACS Nano, 2017. 11: p. 11773-11776.
  4. Ge, C., et al., Towards understanding of nanoparticle-protein corona. Arch Toxicol, 2015. 89: p. 519-39.
  5. Kharazian, B., N.L. Hadipour, and M.R. Ejtehadi, Understanding the nanoparticle-protein corona complexes using computational and experimental methods. International Journal of Biochemistry & Cell Biology, 2016. 75: p. 162-174.
  6. Salvati, A., et al., Transferrin-functionalized nanoparticles lose their targeting capabilities when a biomolecule corona adsorbs on the surface. Nature Nanotechnology, 2013. 8: p. 137-143.
  7. Nel, A.E., et al., Understanding biophysicochemical interactions at the nano-bio interface. Nature Materials, 2009. 8: p. 543-557.
  8. Lunova, M., et al., Nanoparticle core stability and surface functionalization drive the mTOR signaling pathway in hepatocellular cell lines. Sci Rep, 2017. 7: p. 16049.
  9. Feliu, N., et al., In vivo degeneration and the fate of inorganic nanoparticles. Chemical Society Reviews, 2016. 45: p. 2440-2457.
  10. Zhang, Y.N., et al., Nanoparticle-liver interactions: Cellular uptake and hepatobiliary elimination. Journal of Controlled Release, 2016. 240: p. 332-348.
  11. Frtus, A., et al., Analyzing the mechanisms of iron oxide nanoparticles interactions with cells: A road from failure to success in clinical applications. J Control Release, 2020. 328: p. 59-77.

Author Response

Dear reviewer, thank you very much for your interest and positive assessment, as well as for your comments and suggestions for improving the paper.

Round 2

Reviewer 1 Report

The revised version of the manuscript is significantly improved, and suitable for publications in Biomedicines.

Author Response

Большое спасибо!

Reviewer 3 Report

This is very strange reply by the authors. It is admired that they added some additional particle characterization data. However, they completely ignored improving introduction and discussion. Your claims should be backed up with references. The list of references should be increased. The reviewer tentatively proposed only some resent literature. You should improve reference list yourself. Discussion should be elaborated and references should be increased.

Further, protein corona question is not elaborated at all. I remind you what was asked. Authors should comment in detail about protein corona. It becomes evident in nano-bio field that nanomaterials at fist interact with proteins of biological liquids and such interactions change completely desired biological behaviour of nanomaterials. There are numerous studies clearly showing, that when nanoparticles are introduced into biological systems, the surface of nanoparticle immediately interacts with proteins and other biomolecules forming so-called protein corona [1, 2]. The formation of protein corona has important consequences on nanoparticle hydrodynamic diameter, solubility, protein misfolding and aggregation [1-4]. However more importantly, corona also masks the desired biological functionalities [2]. It has been shown that protein corona caused loss of the intended targeting properties of the functionalized nanoparticles [5] and enzymes bound to the surface of nanoparticles change their activity [6, 7]. As a result, it is very hard to make any reasonable conclusion taking into account “naked” nanoparticles. The author’s answer is just word gambling. Do you imply that particles will not interact with serum proteins? Show the data of FTIR, FCS, mass-spec. Alternatively, authors were simply asked to discuss the problem referring to existing literature.

Comment about liver excretion is too loose. You need to relate on previously published data and analysis [8-10] and then propose your hypothesis. Or you mean that your finding is the first in the field? You are making discussion, write explicitly what fits and what does not fit to previously published materials.

Last but not least, highlight comprehensively changes in the manuscript and in point-by-point reply.

  1. Del Pino, P., et al., Protein corona formation around nanoparticles - from the past to the future. Materials Horizons, 2014. 1: p. 301-313.
  2. Ke, P.C., et al., A decade of the protein corona. ACS Nano, 2017. 11: p. 11773-11776.
  3. Ge, C., et al., Towards understanding of nanoparticle-protein corona. Arch Toxicol, 2015. 89: p. 519-39.
  4. Kharazian, B., N.L. Hadipour, and M.R. Ejtehadi, Understanding the nanoparticle-protein corona complexes using computational and experimental methods. International Journal of Biochemistry & Cell Biology, 2016. 75: p. 162-174.
  5. Salvati, A., et al., Transferrin-functionalized nanoparticles lose their targeting capabilities when a biomolecule corona adsorbs on the surface. Nature Nanotechnology, 2013. 8: p. 137-143.
  6. Nel, A.E., et al., Understanding biophysicochemical interactions at the nano-bio interface. Nature Materials, 2009. 8: p. 543-557.
  7. Lunova, M., et al., Nanoparticle core stability and surface functionalization drive the mTOR signaling pathway in hepatocellular cell lines. Sci Rep, 2017. 7: p. 16049.
  8. Feliu, N., et al., In vivo degeneration and the fate of inorganic nanoparticles. Chemical Society Reviews, 2016. 45: p. 2440-2457.
  9. Zhang, Y.N., et al., Nanoparticle-liver interactions: Cellular uptake and hepatobiliary elimination. Journal of Controlled Release, 2016. 240: p. 332-348.
  10. Frtus, A., et al., Analyzing the mechanisms of iron oxide nanoparticles interactions with cells: A road from failure to success in clinical applications. J Control Release, 2020. 328: p. 59-77.

Author Response

(The authors gave the same response as above.)

Round 3

Reviewer 3 Report

Authors have done requested changes.